# Two-Dimensional (PEA)_2_PbBr_4_ Perovskites Sensors for Highly Sensitive Ethanol Vapor Detection

**DOI:** 10.3390/s22218155

**Published:** 2022-10-25

**Authors:** Ching-Ho Tien, Kuan-Lin Lee, Chun-Cheng Tao, Zhan-Qi Lin, Zi-Hao Lin, Lung-Chien Chen

**Affiliations:** 1Department of Electronic Engineering, Lunghwa University of Science and Technology, Taoyuan 33306, Taiwan; 2Department of Electro-Optical Engineering, National Taipei University of Technology, Taipei 10608, Taiwan

**Keywords:** lead halide perovskites, two-dimensional perovskite, (PEA)_2_PbBr_4_, detection, ethanol sensor

## Abstract

Two-dimensional (2D) perovskite have been widely researched for solar cells, light-emitting diodes, photodetectors because of their excellent environmental stability and optoelectronic properties in comparison to three-dimensional (3D) perovskite. In this study, we demonstrate the high response of 2D-(PEA)_2_PbBr_4_ perovskite of the horizontal vapor sensor was outstandingly more superior than 3D-MAPbBr_3_ perovskite. 2D transverse perovskite layer have the large surface-to-volume ratio and reactive surface, with the charge transfer mechanism, which was suitable for vapor sensing and trapping. Thus, 2D perovskite vapor sensors demonstrate the champion current response ratio R of 107.32 under the ethanol vapors, which was much faster than 3D perovskite (R = 2.92).

## 1. Introduction

Vapor sensors are used to detect danger and prevent disasters. They have been applied in industrial sites, laboratories, medical institutions, etc. Owing to symptoms of disease, ethanol may be produced in the human body by the fermentation of intestinal flora or carbohydrate microorganisms in the gastrointestinal (GI) tract. However, the concentration of ethanol exhaled by humans in the presence of disease is very low and is below the resolution limit of current ethanol sensor approaches. Therefore, in this study, we examine the concentration of ethanol exhaled by humans [1,2]. Ethanol is widely used for sensing ethanol concentration after consumption and in various industrial purposes. Therefore, we fabricate vapor sensors for ethanol detection.

Hybrid inorganic-organic metal halide perovskites have received considerable attention in the applications of solar cells [3], light-emitting diodes (LED) [4], photodetectors [5], and lasers [6]. For perovskite solar cells, the power conversion efficiency (PCE) of perovskite solar cells has increased from 3.8% to 25.7% [7,8] in the past decade, which was very close to traditional silicon-based solar cells. Recently, perovskite-based LEDs have also achieved a significant external quantum efficiency of more than 20% [9]. It was worth noting that organic-inorganic hybrid halide perovskite materials have attracted much attention in gas sensing due to their high environmental sensitivity and room-temperature high carrier mobility [10,11]. Recently, various perovskite-based sensors have been reported for the detection of humidity [12], O_2_ [13], NO_2_ [14,15] and NH_3_ [16,17,18], etc. When a perovskite film is exposed to vapor molecules, the vapor molecules may fill the vacancies in the perovskite film, significantly increasing the film conductivity. In contrast, when inert vapors reach the perovskite film, the inert vapor molecules remove the target vapor molecules, increasing the number of vacancies, which decreases the perovskite film conductivity [13,14,19].

Although perovskites have made significant breakthroughs in device efficiency, three-dimensional (3D) perovskites (especially MA-based perovskites) were prone to degradation when exposed to water, oxygen, heat and continuous light due to their poor stability [20,21,22,23], therefore, the optoelectronic applications and long-term stability were limited. In contrast, the emerging two-dimensional (2D) layered perovskite has received more attention due to its good stability, unique structure and excellent optoelectronic properties [24,25,26]. Because the phenylethylamine cation (PEA^+^) as the organic cation on the surface of 2D Ruddlesden-Popper (RP) phase perovskite have moisture resistance and hydrophobicity and van der Waals force in the structure, 2D RP phase perovskite has higher structural stability than 3D perovskite [27,28,29]. Duong et al. reported the use of the bulky organic cation n-octylammonium bromide to passivate 3D perovskites, resulting in mixed-dimensional 2D/3D perovskites. The 2D/3D perovskite gas sensor based on detecting NO_2_ gas exhibits excellent performance and ambient stability, with a sensitivity of 6.3 ± 0.83 times per ppm and fast response/recovery times of 5.7 s and 12.7 s, respectively [30]. Adli et al. reported the synthesis of lead iodide-based and lead nitrate-based 2D aminoethyl methacrylate perovskites for ammonia gas sensor via a one-step sequential reaction. 2D layered hybrid perovskites based on lead nitrate enhance the stability of NH_3_ sensing compared to lead iodide [31]. These findings highlight the potential of using 2D perovskites to produce gas sensors. Based on the above reasons, we turned our attention to the 2D RP (PEA)_2_PbBr_4_ perovskites (PEA(C_6_H_5_C_2_H_4_NH_3_): phenethylammonium). It was expected that compared with common MA^+^ cations, the presence of PEA^+^ cations could enhance the hydrophobicity and stability of perovskites. In this work, the manufactured (PEA)_2_PbBr_4_ based vapor sensor showed relatively good sensing performance when exposed to ethanol vapors.

## 2. Materials and Methods

### 2.1. Materials

Lead (II) bromide (PbBr_2_, 99.998%), Methylammonium bromide (MABr, 98%), and phenethylammonium bromide (PEABr, ≥98%) were purchased from Sigma-Aldrich (Sigma-Aldrich, St. Louis, MO, USA). N,N-dimethyl formamide (DMF, 99.8%) was purchased from Uni-Onward (New Taipei, Taiwan). Patterned ITO-coated glass substrate (7 Ω sq^−1^) was purchased from Ruilong (New Taipei, Taiwan).

### 2.2. Sensor Fabrication

Two perovskite solutions were used in this study; a 3D and a 2D solution. For the 3D solution, PbBr_2_ was mixed with MABr in a ratio of 1:1 mol, then dissolved in 1 mL DMF. For the 2D solution, PbBr_2_ was mixed with 2D PEABr in a ratio of 1:2 mol, then dissolved in 1 mL DMF. The prepared perovskite solution (60 μL) was deposited on the ITO conductive glass, and first spun at 1000 rpm for 40 s using a spin coater. Subsequently, the 2D-(PEA)_2_PbBr_4_ perovskite films were formed by spinning at 5500, 6500, and 7500 rpm, respectively, for 20 s. In the last 10 s, 200 μL of toluene was deposited on the surface and heated on a heating plate at 100 °C for 10 min to form a 2D-(PEA)_2_PbBr_4_ perovskite film. Figure 1a shows the crystal structure of (PEA)_2_PbBr_4_. The -NH_2_ group from the PEA cations were bonded to the [PbBr*_x_*]^2−*x*^ octahedron, which assembles into the 2D structure. Figure 1b shows the schematic diagram of the (PEA)_2_PbBr_4_ alcohol vapor sensor. The experimental setup used to characterize the sensing performance of the vapor sensor, the measurement process was carried out at room temperature, as shown in Figure 1c.

### 2.3. Measurements

#### 2.3.1. Current–Voltage (I–V) Curve Measurement

The perovskite sensor was placed in a self-designed container and voltages from 0 to 4 V were applied to record their original I–V characteristics at room temperature (25 ± 3 °C). Subsequently, the sensor was placed in an environment filled with ethanol vapor (10,000 ppm), and the relative humidity was 25–30%. Before characterization, the air in the container was completely vented with high-purity N_2_ gas. Afterwards, a specific volume of ethanol vapors was injected into the container, followed by N_2_ injection immediately after the ethanol vapors was stopped. Ethanol vapor was generated from liquid ethanol-water mixture using an ultrasonic nebulizer and then injected into a Tedlar bag. The desired ethanol vapor concentration was adjusted by appropriately diluting the liquid ethanol with deionized water. After 30 s, the sensors were removed and the same voltages (0–4 V) were applied to record the changes in the I–V characteristics. At 15 and 30 s after removal, the I–V characteristics of the sensors were recorded to observe the restored states of the sensors and compare them with the original I–V curves. Hence, the change in the current of the perovskite film and its recoverability after the exposure to ethanol vapor were obtained.

#### 2.3.2. Current–Time (I–T) Curve Measurement

To observe the sensitivities of the sensors over a period, the I–T characteristics at room temperature of the perovskite sensors were measured. A voltage of 4 V was applied and a period of 60 s was used as the cycle. In the first cycle, the sensors were placed in the air. In the second cycle, the sensors were placed in an environment filled with ethanol vapor. Subsequently, odd cycles followed the same operation as the first cycle, and even cycles followed the same operation as the second cycle. In total, five cycles were recorded to observe the corresponding I–T characteristics.

#### 2.3.3. Responsivity Measurement

Owing to the difference between the original resistance of each conductive glass, the responsivities were compared with the measured I–V characteristics. In this method, the kinetic current measured after adding the target vapor (*I*) was subtracted from the original current measured in the absence of vapor (*I*_0_) to obtain the current response caused by the thin film. Subsequently, the difference was divided by the original current (*I*_0_), which yielded the normalized current response ratio (*R*) of the sensor. The equation is expressed as follows:(1)$$R=\frac{\Delta I}{I_{0}}=\frac{I-I_{0}}{I_{0}}$$

#### 2.3.4. Photoluminescence (PL) Measurement

The crystal defect density and crystallinity of the perovskite films can be studied by PL measurement, and their correlations with the film responses can be confirmed. The PL was measured using a HITACHI F-7000 (Hitachi, Tokyo, Japan) with a 325-nm excitation light. The perovskite films were placed into the instrument for testing. For data acquisition, the films were placed in an ethanol environment for 30 s and then removed; immediately, they were placed into the PL measuring instrument for measurement.

## 3. Results and Discussion

The conductive glass used in this study was ITO (indium tin oxide) on glass. The images of the 3D and 2D perovskite films were shown in Figure 2. An apparent color difference can be observed between the 2D-(PEA)_2_PbBr_4_ and 3D-MAPbBr_3_ perovskite films. The 3D perovskite appeared as orange films, and the 2D perovskite appeared as white films. Next, the sensitivities of 3D and 2D perovskites were discussed and the influence of different thicknesses on the film conductivities were studied.

The responses of 3D-6500rpm-MAPbBr_3_ and 2D-6500rpm-(PEA)_2_PbBr_4_ exposed to air were compared with those exposed to 95% (v/v) ethanol vapor. When the perovskite thin film was exposed to ethanol vapor, the vapor molecules filled the vacancies in the film, significantly increasing the conductivity. When the perovskite film was removed from the ethanol vapor environment, inert vapor molecules (N_2_ gas) oved the ethanol vapor molecules from the film, forming new vacancies, and decreased the perovskite film conductivity. Figure 3 shows that 2D-6500rpm-(PEA)_2_PbBr_4_ exhibited a better response than 3D-6500rpm-MAPbBr_3_. From the responsivity calculation, it can be observed that the 2D-6500rpm-(PEA)_2_PbBr_4_ thin film exhibited a 37 times higher response for 95% (v/v) ethanol vapor sensing than the 3D-6500rpm-MAPbBr_3_ thin film.

To investigate the relationships between the fabricated film structures and their sensitivities as well as identify the differences in their structures, the two perovskite films were observed using a field-emission scanning electron microscope (FESEM, Sigma, ZEISS, Munich, Germany). Figure 4a,b show the low magnification SEM images of the 3D-6500rpm-MAPbBr_3_ and 2D-6500rpm-(PEA)_2_PbBr_4_ perovskites. Although both materials had no large pores, large crystals were clearly observed on the 3D-6500rpm-MAPbBr_3_ perovskite, whereas most of the films fabricated using the 2D-6500rpm-(PEA)_2_PbBr_4_ perovskite showed an evenly distributed surface. Considering the actual material structures, the 2D-6500rpm-(PEA)_2_PbBr_4_ perovskite has an even and fine surface, and the area for vapor sensing is relatively uniform and consistent. Such a thin-film structure has a higher surface to volume ratio and therefore has a faster response time. Therefore, the 2D-6500rpm-(PEA)_2_PbBr_4_ perovskite may be more suitable for vapor sensing than the 3D-6500rpm-MAPbBr_3_ perovskite. Figure 4c,d show the 10 KX high magnification SEM images of the 3D-6500rpm-MAPbBr_3_ and 2D-6500rpm-(PEA)_2_PbBr_4_ perovskites, in detail. The 3D-6500rpm-MAPbBr_3_ perovskite film could clearly identify crystals of different sizes and uneven surfaces. In contrast, the 2D-6500rpm-(PEA)_2_PbBr_4_ film structure exhibited dense and uniformly distributed fine pores. Thus, the decrease in the responsivity of the 3D-6500rpm-MAPbBr_3_ perovskite film is attributed to the uneven surface and the presence of large crystals. Furthermore, as the 2D-6500rpm-(PEA)_2_PbBr_4_ film had a uniformly distributed and dense surface, it exhibited better sensitivity and conductivity.

Figure 5a shows the comparison of the PL intensity of 3D-6500rpm-MAPbBr_3_ and 2D-6500rpm-(PEA)_2_PbBr_4_ perovskites. Notably, the PL intensity of the 2D-6500rpm-(PEA)_2_PbBr_4_ perovskite was much larger than that of the 3D-6500rpm-MAPbBr_3_ perovskite. This further proves that the crystal defect density of the 2D-6500rpm-(PEA)_2_PbBr_4_ perovskite was much lower than that 3D-6500rpm-MAPbBr_3_ perovskite, leading to better conductivity. In addition, the PL emission peak of 3D-6500rpm-MAPbBr_3_ was located at about 550 nm, and the 2D-6500rpm-(PEA)_2_PbBr_4_ was at about 415.2 nm. Investigating the crystal structure was characterized by X-ray diffractometer (X’Pert PRO MRD, PANalytical, Netherlands) with CuKα (λ = 1.5418 Å) radiation source. Figure 5b shows the X-ray diffraction (XRD) patterns for 3D-6500rpm-MAPbBr_3_ and 2D-6500rpm-(PEA)_2_PbBr_4_ perovskites. the characteristic peaks appearing at 5.19°, 10.52°, 15.94°, 21.31°, 26.73°, 32.22°, and 37.7° on the 2D-6500rpm-(PEA)_2_PbBr_4_ perovskite pattern, which are ascribed to (002), (004), (006), (008), (010), (012), and (014) planes, respectively. The peaks at 14.96°, 21.2°, 30.16°, 33.85°, 37.16°, 43.23°, and 45.89° on the 3D-6500rpm-MAPbBr_3_ perovskite pattern represent the (100), (110), (002), (210), (212), (220), and (003) planes, respectively. It was found that the lattice planes of the 2D-6500rpm-(PEA)_2_PbBr_4_ perovskite were better than those of the 3D-6500rpm-MAPbBr_3_ perovskite. Among them, the (002) plane exhibited good crystallinity and a high degree of preferred orientation.

From the above results, it was found that 2D-6500rpm-(PEA)_2_PbBr_4_ showed a better response to the vapor than 3D-6500rpm-MAPbBr_3_. This result was also proved by the SEM, PL, and XRD observations. The crystal structure of 2D-6500rpm-(PEA)_2_PbBr_4_ was of good quality, the distribution was finer and more uniform, which increased the reaction area of the film. In contrast, the 3D-6500rpm-MAPbBr_3_ perovskite had relatively large crystal and thin film inhomogeneities, resulting in poor performance for vapor sensing. In order to improve the sensitivity of 2D-(PEA)_2_PbBr_4_ perovskite vapor sensing, we further explored the effect of 2D-(PEA)_2_PbBr_4_ perovskite vapor sensing under different spin coating conditions.

Optical absorption spectra were conducted using a UV-Visible-Near Infrared (UV/VIS/NIR) spectrophotometer (V-770, Jasco, Japan), as shown in Figure 6a, the samples of 2D-(PEA)_2_PbBr_4_ perovskite for various spin coating speeds showed an absorption edge at 420 nm with an energy gap of 2.95 eV, as shown in Figure 6b. It does consistent with the PL spectrum as show in Figure 5a. The electron energy states of crystal do not change due to various film thickness and spin coating speeds.

The top view FESEM images of the 2D-(PEA)_2_PbBr_4_ films formed with different spin speeds were shown in Figure 7a–c. The 2D-(PEA)_2_PbBr_4_ formed at 7500 rpm shows the most disordered structure among the three. This was possibly because the grain boundaries may not have been smoothly formed with the high spin speed. In terms of crystal structure, the film formed at 5500 rpm had more grain boundaries than that formed at 6500 rpm. When the spin speed was low, a slightly defective structure formed. The thin film formed at 6500 rpm showed the best structure, which had fewer grain boundaries, exhibited higher uniformity, and had a better crystal structure. A comparison of the films formed at 7500 and 5500 rpm showed that the former performed slightly better in terms of the vapor contact and conductivity, which can be attributed to its larger contact area. In addition, Figure 7e–f were cross-section SEM images of 2D-(PEA)_2_PbBr_4_ films made at different spin speeds. It can be seen that the 2D-(PEA)_2_PbBr_4_ film was squeezed together at 7500 rpm due to the excessive speed of the speed, causing the crystal structure to be chaotic. Overall, the 2D-(PEA)_2_PbBr_4_ perovskites distribution at 5500 rpm and 6500 rpm were relatively even, and the crystal structure and grain boundary size at 6500 rpm was better than those at 5500 rpm and 7500 rpm.

Figure 8 shows the XRD patterns of 2D-(PEA)_2_PbBr_4_ films formed with different spin speeds. It can be observed that the crystal lattice plane of 2D-6500rpm-(PEA)_2_PbBr_4_ film was significantly better than the other two. Although the three have the best response on the (002) crystal plane, the 2D-6500rpm-(PEA)_2_PbBr_4_ film has good crystallinity and high degree of preference orientation.

During the fabrication of the perovskite films, high-density crystal defects were produced on the surface owing to the evaporation of organic solvent during annealing. These defects hindered the movement of electrons in the conductive band. Thus, the perovskite thin film exhibited low conductivity. However, after the perovskite film absorbed the target vapor, these defects passivated and the electrons trapped in the defects were released into the conductive band, increasing the film conductivity. After absorbing the N_2_ inert vapor, the specific vapor was discharged, and the crystal defects formed again on the film. This was proved by the PL measurements. The graph in Figure 9 shows that the measured PL intensities before the exposure to ethanol vapors were all lower than those after the exposure to ethanol vapors. This was because after the exposure to ethanol vapors, the density of crystal defects decreased, increasing the crystallinity. Thus, the PL intensities also increased. The PL intensities measured for the 2D-(PEA)_2_PbBr_4_ films formed at different spin speeds were shown in Figure 9. After the exposure to ethanol vapors, the PL intensities slightly increased. However, as the 2D-(PEA)_2_PbBr_4_ film thickness changed, the PL intensities varied. The PL intensity of the 2D-(PEA)_2_PbBr_4_ film formed at 6500 rpm was the highest. Thus, the corresponding crystal structure was denser and more uniform, which was consistent with the I–V measurements.

The current response ratio for different spin coating speeds of the 2D-(PEA)_2_PbBr_4_ films were investigated. From the FESEM results, it was known that the 2D-6500rpm-(PEA)_2_PbBr_4_ perovskite film was relatively dense and had fewer grain boundaries. The 2D-5500rpm-(PEA)_2_PbBr_4_ perovskite was relatively uniform, but there were many grain boundaries. The I–V curves of the (PEA)_2_PbBr_4_ films formed with three different spin speeds in 95% (v/v) ethanol vapor were shown in Figure 10a, and the responses were calculated as shown in Figure 10b. The highest response ratio of the 2D-6500rpm-(PEA)_2_PbBr_4_ perovskite film (107.32) was much higher than that of the other two 2D-5500rpm-(PEA)_2_PbBr_4_ perovskite (62.54) and 2D-7500rpm-(PEA)_2_PbBr_4_ perovskite (48.58). This was possibly because when the film had fewer grain boundaries, the carrier transmission efficiency was higher, and the response to the chemical substance in the target vapor was stronger. On the other hand, the 2D-7500rpm-(PEA)_2_PbBr_4_ perovskite has a lower response, which may be attributed to the presence of more grain boundaries in the film that affected the response.

The response times of the 2D-(PEA)_2_PbBr_4_ perovskites formed with different spin speeds in 95% (v/v) ethanol vapor were calculated, as shown in Figure 11a. The 2D-6500rpm-(PEA)_2_PbBr_4_ film showed a continuous response to ethanol vapor during four cycles due to its good uniform density and crystal quality, confirming that exposure to ethanol vapor seems not to damage the structural integrity of 2D-6500rpm-(PEA)_2_PbBr_4_ film. Overall, the 2D-6500rpm-(PEA)_2_PbBr_4_ film exhibits a response time of 8.83 s and a recovery time of 0.43 s, which was faster than the response times (5.85 and 9.85 s) and recovery times (0.86 and 0.44 s) of the 2D-5500rpm-(PEA)_2_PbBr_4_ and 2D-7500rpm-(PEA)_2_PbBr_4_, respectively. Among the three, the 2D-5500rpm-(PEA)_2_PbBr_4_ film exhibited the worst performance. In the first cycle in Figure 11a, its response time was quicker than the 2D-7500rpm-(PEA)_2_PbBr_4_ film. However, from the second cycle test, its response time slowed down, and its response gradually declined. For the 2D-5500rpm-(PEA)_2_PbBr_4_ film, the current decreases with time, and the trend in the first cycle was found to be almost double that observed in the fourth cycle. It was considered that the excessive number of grain boundaries caused a relatively poor carrier transmission efficiency, which led to the gradual decline in its response. To verify the sensitivity of the best 2D-6500rpm-(PEA)_2_PbBr_4_ vapor sensor, different ethanol vapor concentrations (35%, 55%, 75%, and 95% v/v) were measured. Figure 11b shows that the various ethanol vapor concentration has a linear relationship with the sensing current. The sensing current was lowest at 35% ethanol vapor concentration. As the ethanol concentration increased, the sensing current increased accordingly. After calculation, the current response ratios (@3.9V) of the 2D-6500rpm-(PEA)_2_PbBr_4_ vapor sensor at 35%, 55%, 75%, and 95% v/v ethanol vapors were 25.74, 32.29, 37.23, and 53.81, respectively.

A schematic of the gas-sensing mechanism of the 2D-(PEA)_2_PbBr_4_ vapor sensor was shown in Figure 12. When the 2D-(PEA)_2_PbBr_4_ vapor sensor was exposed to fresh air; oxygen molecules (O_2_) tend to adsorb on the surface of 2D-(PEA)_2_PbBr_4_ and capture electrons to generate various chemisorbed oxygen ions (such as O^−^, O^2−^, or O_2_^−^) [32,33]. The chemisorbed oxygen creates a high surface potential barrier on the surface of the 2D-(PEA)_2_PbBr_4_; leading to the formation of a wide electron depletion layer; thereby increasing the resistance of 2D-(PEA)_2_PbBr_4_. When the 2D-(PEA)_2_PbBr_4_ vapor sensor was exposed to ethanol vapor; the adsorbed ethanol molecules react with the chemisorbed oxygen species on the surface of 2D-(PEA)_2_PbBr_4_ to form CO_2_ and H_2_O; resulting in the chemisorbed electrons that will freely return to 2D-(PEA)_2_PbBr_4_; increasing the conductivity. Thus; the amounts of oxygen adsorbed on the surface of 2D-(PEA)_2_PbBr_4_ decreases; the width of the electron depletion layer was narrowed and the surface potential barrier was lowered; resulting in low resistance of 2D-(PEA)_2_PbBr_4_

A comparative table of the most recent ethanol sensing perovskites candidates was summarized in Table 1. Most ABO_3_-based perovskite metal oxides gas sensors respond to ethanol at high temperatures. Here, the response of the 2D-(PEA)_2_PbBr_4_ perovskite reaches approximately 107.32 toward 95% v/v ethanol vapors at room temperature. The 2D-(PEA)_2_PbBr_4_ perovskite sensor exhibits an excellent performance and room temperature-based operating temperature compared to other ABO_3_-type sensors reported in the literature, providing a potential application for ethanol detection.

## 4. Conclusions

In summary, the perovskite film fabricated using 2D-(PEA)_2_PbBr_4_ for sensing volatile organic vapor exhibited a good response to the target vapor and recoverability at room temperature. It did not require any external signals to be restored and can serve as a good sensor in the future. The presence of the target vapor directly affected the conductivity of the perovskite thin film and varied it. A charge trap mechanism was employed: once the perovskite thin film absorbed the target vapor, its crystal defects were repaired, thereby releasing the trapped electrons and improving the thin film conductivity. Compared with 3D-MAPbBr_3_ perovskite, 2D-(PEA)_2_PbBr_4_ perovskite film has the best crystalline quality and uniform dense film at 6500 rpm, which significantly optimizes sensor performance. In the future, 2D perovskite film can be fabricated to develop stable vapor sensor with low power consumption, low cost, and wide applicability in various fields.

## Figures and Tables

**Figure 1 sensors-22-08155-f001:**
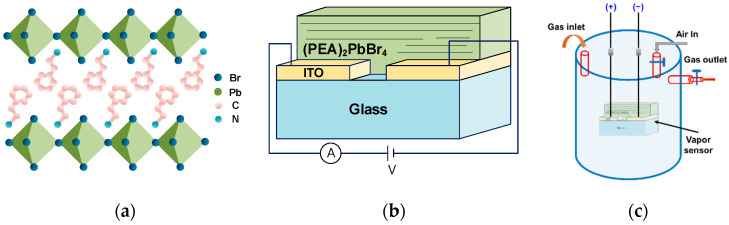
(**a**) Crystal structure of 2D-(PEA)_2_PbBr_4_; (**b**) Schematic diagram of the 2D-(PEA)_2_PbBr_4_ alcohol vapor sensor; (**c**) Gas sensing measurement setup schematic.

**Figure 2 sensors-22-08155-f002:**
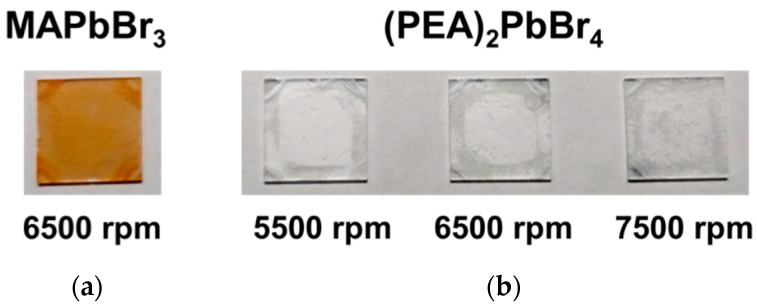
3D and 2D perovskites sensing samples. (**a**) 3D MAPbBr_3_, (**b**) 2D (PEA)_2_PbBr_4_ with different spin speeds.

**Figure 3 sensors-22-08155-f003:**
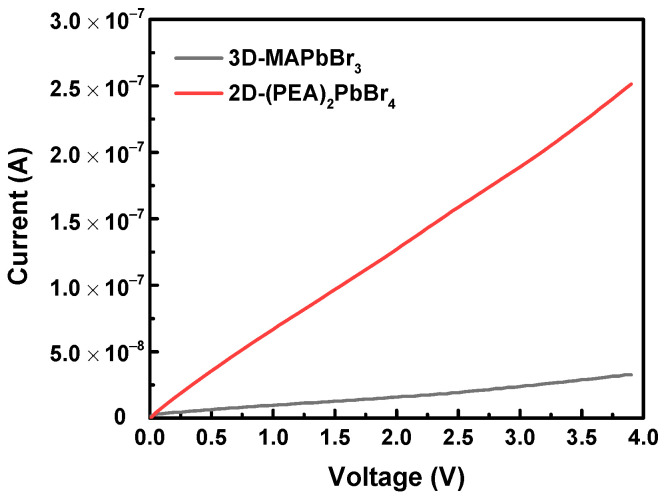
Current–voltage (I–V) curves of 3D-6500rpm-MAPbBr_3_ and 2D-6500rpm-(PEA)_2_PbBr_4_ perovskites after exposure to 95% (v/v) ethanol vapor.

**Figure 4 sensors-22-08155-f004:**
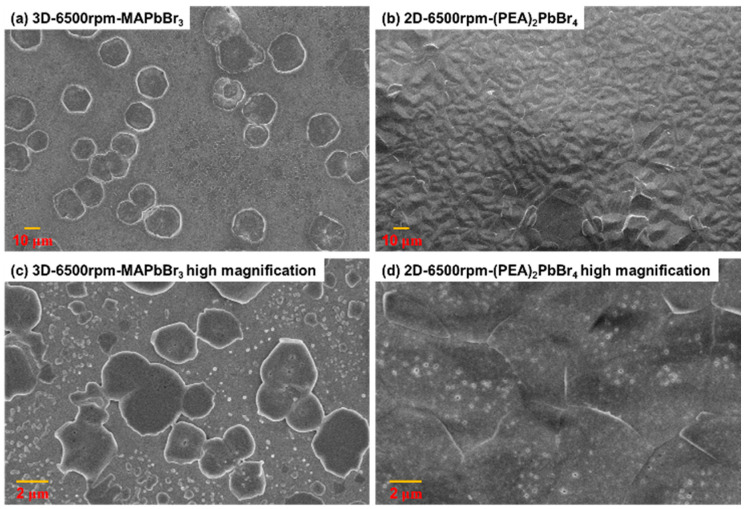
Low and high magnification field-emission scanning electron microscope (FESEM) images of (**a**,**c**) 3D-6500rpm-MAPbBr_3_ and (**b**,**d**) 2D-6500rpm-(PEA)_2_PbBr_4_ perovskites.

**Figure 5 sensors-22-08155-f005:**
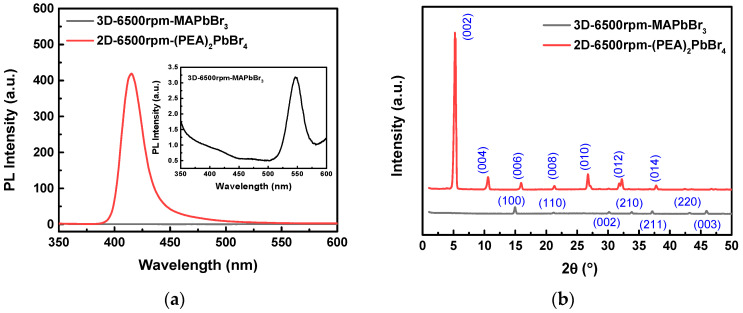
Comparison of (**a**) photoluminescence (PL) intensities and (**b**) X-ray diffraction (XRD) patterns of 3D-6500rpm-MAPbBr_3_ and 2D-6500rpm-(PEA)_2_PbBr_4_ perovskites.

**Figure 6 sensors-22-08155-f006:**
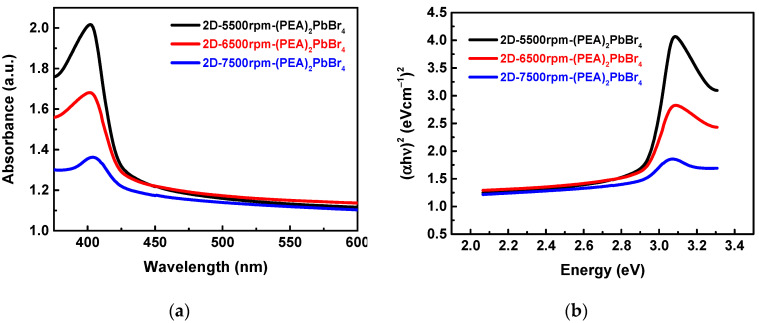
(**a**) Optical absorption spectra and (**b**) (αhν)^2^ vs. energy plot of 2D-(PEA)_2_PbBr_4_ with different spin speeds.

**Figure 7 sensors-22-08155-f007:**
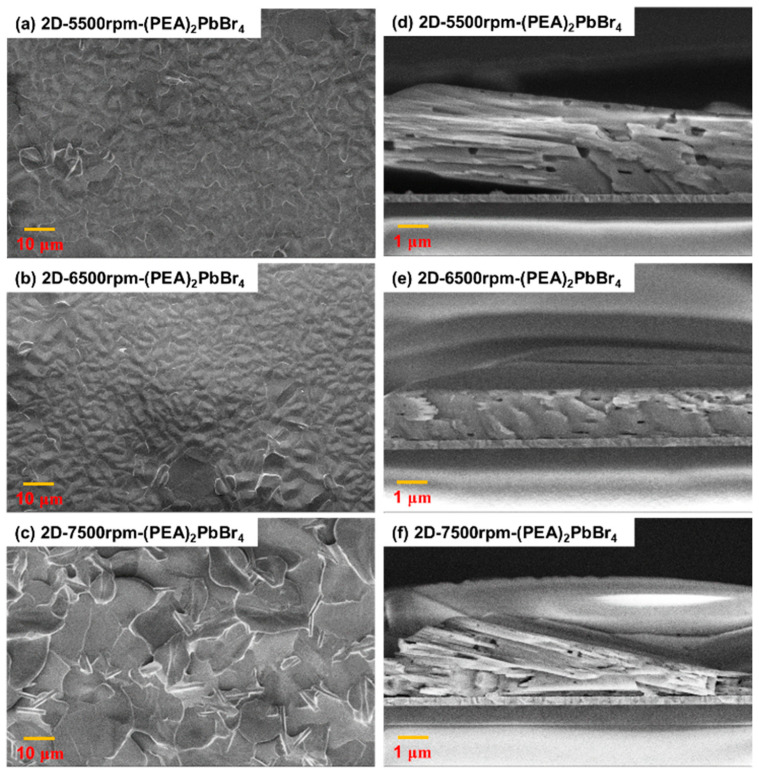
(**a**–**c**) Top view and (**d**–**f**) cross-section field-emission scanning electron microscope (FESEM) images of 2D-(PEA)_2_PbBr_4_ films formed with different spin speeds.

**Figure 8 sensors-22-08155-f008:**
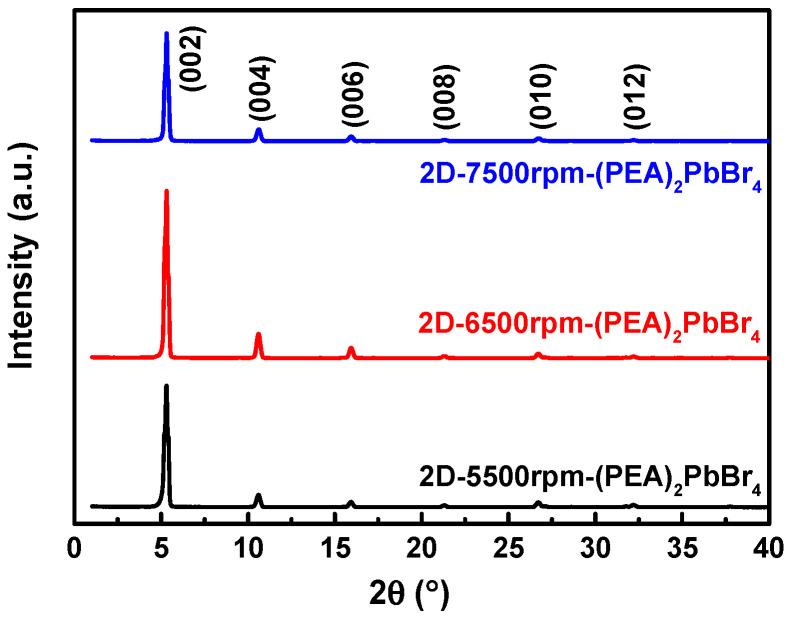
X-ray diffraction (XRD) patterns of 2D-(PEA)_2_PbBr_4_ films formed with different spin speeds.

**Figure 9 sensors-22-08155-f009:**
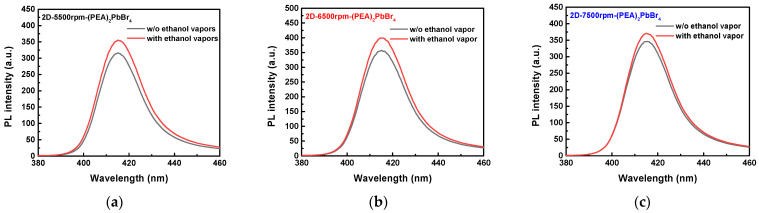
Photoluminescence (PL) intensities of 2D-(PEA)_2_PbBr_4_ films formed with different spin speeds in ethanol vapors environment. (**a**) 2D-5500rpm-(PEA)_2_PbBr_4_ film. (**b**) 2D-6500rpm-(PEA)_2_PbBr_4_ film. (**c**) 2D-7500rpm-(PEA)_2_PbBr_4_ film.

**Figure 10 sensors-22-08155-f010:**
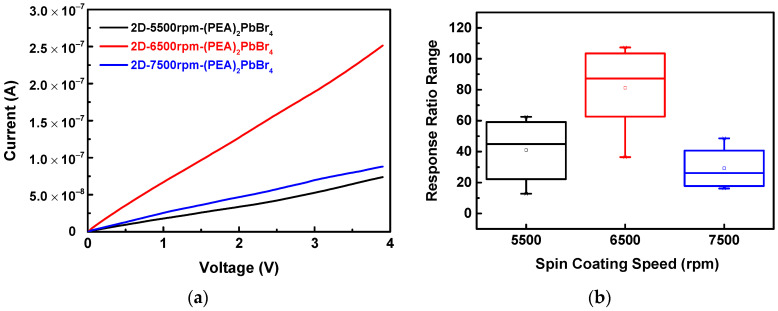
(**a**) Current–voltage (I–V) curves and (**b**) response ratio ranges of 2D-(PEA)_2_PbBr_4_ films formed at different spin coating speeds to ethanol vapors.

**Figure 11 sensors-22-08155-f011:**
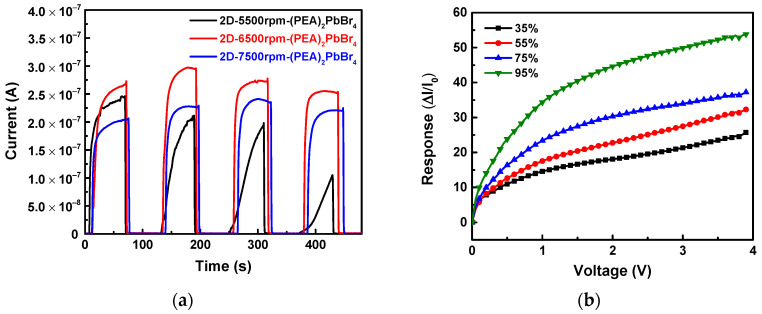
(**a**) Current–time (I–T) curves of 2D-(PEA)_2_PbBr_4_ films formed with different spin speeds in ethanol vapors environment. (**b**) Sensor response of 2D-6500rpm-(PEA)_2_PbBr_4_ vapor sensor after exposure to different concentrations of ethanol vapors.

**Figure 12 sensors-22-08155-f012:**
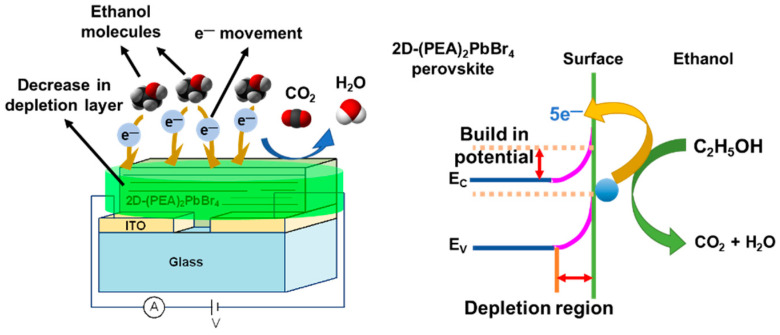
Schematic ethanol vapors sensing mechanism of 2D-(PEA)_2_PbBr_4_ perovskite.

**Table 1 sensors-22-08155-t001:** Comparative summary of the ethanol sensing performance of the most recent various perovskite-based sensors.

Material	Operating Temperature (°C)	Concentration (ppm)	Response	Response/Recovery Time (s)	Year/Ref.
Ag/LaFeO_3_	180	20	21	26/27	2022/[34]
ZnSnO_3_ porous bodies	270	80	37	4/581	2022/[35]
BaSnO_3_ porous bodies	270	80	14.3	72/596	2022/[35]
Ba_0.9_La_0.1_FeO_3_	250	1	143	--	2022/[36]
Hollow LaFeO_3_	300	143	14.5	23/39	2022/[37]
Ag–ZnSnO_3_	200	100	83.9	1/50	2021/[38]
Ag-LaFeO_3_	190	100	155	30/5	2021/[39]
Cs_3_Bi_2_I_6_Br_3_	RT	500	3.7	--	2022/[40]
(PEA)_2_PbBr_4_	RT	10000	107.32	8.83/0.43	This work

## Data Availability

Not applicable.

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
