# Peer review of "Two-Dimensional (PEA)2PbBr4 Perovskites Sensors for Highly Sensitive Ethanol Vapor Detection"

_sensors, 2022, doi:10.3390/s22218155_

Round 1

Reviewer 1 Report

The authors report on the detection of ethanol vapor using Two-dimensional (PEA)2PbBr4 perovskites based gas sensors.

The articles is well-written and focused. The results are clear discussed.

In order to increase the impact of the article the authors must suggest a model for the sensing mechanisM

Author Response

See attached file, please.

Reviewer 2 Report

Author Ching-Ho Tien et. al reported “Two-dimensional (PEA)2PbBr4 perovskites sensors for highly sensitive ethanol vapor detection”. The author reported that the 2D-(PEA)2PbBr4 perovskite of the horizontal vapor sensor was outstandingly superior to 3D-MAPbBr3 perovskite by using the transverse perovskite layer of the 2D perovskite. Also, the author reported 2D perovskite vapor sensors with a current response ratio R of 107.32 under the ethanol vapors better than 3D perovskite (R=2.92). The authors have explained their work well and the figures were arranged well. But this current work needs some revision before publishing in the Sensors MDPI journal.

1. The author should check Lines 65 and 198.

2. In figure 1, the author has given (a) and (b) in the figure caption but did not mention them in the figures.

3. The author should check the figure 7 caption.

4. The author must use the word “Figure” instead of “Fig.” throughout the manuscript.

5. In figure 13. The various concentration of ethanol vapors at 0 V starts from various current. Why is this change current?

6. The author should check all the figures and figure captions in the manuscript.

Author Response

See attached file, please.

Reviewer 3 Report

The authors investigate a perovskite film-based approach for ethanol vapor detection. The manuscript is clear and understandable with good illustrations. The reproducibility of the experiments is partially given, i.e., the measurements of ethanol vapor lack information on the atmospheric conditions and also how the atmosphere was controlled and stabilized. In particular, I am concerned that studies on the dependence of the sensor on humidity are missing. The humidity could have influenced the measurements. This should be clarified and supported with measurements. Furthermore, a comparison of the sensitivity and selectivity of the investigated perovskite film sensor with other sensor approaches for the measurement of ethanol vapor would be recommended. This also, among other reasons, because the authors refer to highly sensitive ethanol vapor in the title of the manuscript. Before publication, the issues listed below should be resolved.  

 1. Line 28, the expression "negligible" refers to a reference quantity, the sentence should be rephrased, e.g., "the concentration of ethanol exhaled by humans in the presence of disease is very low and is below the resolution limit of current ethanol sensor approaches.”

2.      Line 65, typo Figure 1a

3.     Line 87 and following: The sensor was tested in air as a reference (baseline). However, firstly, it is unclear what the atmosphere "air" consisted of, e.g. humidity, temperature. Second, the sensitivity of the sensor to humidity was not investigated. Reference 10 shows that perovskite sensors are sensitive to humidity.  The dependence on humidity should be investigated and measured. For example, the measurements in the ethanol vapor atmosphere may have been made under different (higher ??) humidity conditions than the reference measurements in air. It should be clear to what extent the ethanol sensitivities obtained were affected by humidity, and if there is a dependence on humidity, it would be desirable if this effect could be compensated to show the dependencies on ethanol only (Figure 3, 11 and 12).

4.      The temperature must be specified for all measurements (currently missing)

5.      Line 89 and onward. What does it mean that the sensor was placed in an environment filled with ethanol vapor? Was the ethanol vapor pressure equal to the saturation vapor pressure at the temperature in the measurement chamber? It would be desirable if the measurement chamber was shown with an illustration. It is also unclear how the ethanol vapor concentration was controlled and stabilized. What were the other atmospheric conditions, such as humidity and other gases?

6.      Line 99, as point 5.

7.      Line 123, suggested change: “the sensitivities of 3D and 2D perovskites are discussed and the influence of different thicknesses on the film conductivities are studied.”

8.      Line 132, what were the inert vapor molecules? This also goes into the question of the atmosphere in the measurements (see points 3, 5)

9.      Line 139, figure 3, the ethanol vapor concentration should be specified

10.   Line 150, I would say the thin films have a higher surface to volume ratio and therefore have a faster response time

11.   Line 191 - 193, the explanations in the text are unclear.

12.   Line 200, Figure 7 shows measurements from 2D and not from 3D samples. The caption must be corrected.

13.   Line 261-262, The reaction time of the 2D (PEA)2PbBr4 perovskite fabricated with different spins was inferred from the measurements shown in Figure 12

14.   Line 267-268, please rephrase, this is unclear: “For the 2D-5500rpm-(PEA)2PbBr4 267 film, the first increasing trend was almost double that observed in the fourth cycle”.

15.   Line 274-276, please rephrase to correct grammar.

16.   Line 275, It is unclear what the percentages refer to? Is this the saturation pressure? See also point 5.

17.   It would be useful to compare the sensitivity/resolution possibly selectivity ethanol versus humidity of the investigated perovskite sensor with other sensor approaches e.g. MOX technology. This helps to classify the perovskite approach.

Author Response

See attached file, please.

Reviewer 4 Report

This report presents two-dimensional (2D) 2D-(PEA)2PbBr4 perovskite for sensor application. It is claimed to have better vapor-sensing properties than 3D-MAPbBr3 perovskite. They have compiled a report with a set of related characteristics and insights. There is much room to improve  data presentation.  It is can be considered for publication after major revision.

Comments:

1.     4th paragraph of the introduction seems to be mixed up. It is suggested to revise the content focusing on the outputs of this work.

2.     In the experimental section, please include the chemical company name. XRD measurement information is also missing.

3.     PL response of the films is given in Fig. 10.  Can electroluminescence response be measured as reported in another report?  (https://pubs.rsc.org/en/content/articlelanding/2017/TC/C7TC02822A; https://pubs.acs.org/doi/10.1021/acs.jpclett.0c02363). It is recommended to discuss these aspects to widen the scope of this report.

4.     In Fig. 11, it has mentioned the rpm speed and properties. Can it be correlated with some properties such as thickness, film roughness, or grain size of the 2D film rather than rpm speed? It would be more scientific.

5.     It would be better if Fig. 5 and Fig. 6 are pleased together. The same action can be done for Fig. 12 and 13.

Author Response

See attached file, please.

Round 2

Reviewer 3 Report

Most of the issues have been addressed in the revision. Before publication, the issues below should be addressed, a few of which have not been resolved since the last review.

 1.      Figure 1a, 1b not visible

22.      Line 102-103: The sentence should be improved for clarity, for example: “Ethanol vapor was generated from liquid ethanol-water mixture using an ultrasonic nebulizer and then injected into a Tedlar bag. The desired ethanol vapor concentration was adjusted by appropriately diluting the liquid ethanol with deionized water.”

33.      The relative percent ethanol concentration is not clear, what is the reference (100%)? For example, the saturation vapor pressure at the experimental temperature. This is the same as point 16 in the previous review.

44.      Line 296-297: the voltage for the specified current response ratios should be specified (probably at 4V?).

55.      line 301..313 and figure 12: This is a helpful description of the chemical reaction at the surface of the perovskite sensor, but in the abstract the sentence “The transverse perovskite layer could confine carrier in one-dimensional horizontal channels to accomplish effective high-speed carrier transport for the horizontal vapor sensor” does not fit to that. So, the abstract should be corrected accordingly.

66.      Line 310: …electrons that will freely return…

77.      Previous review point 17: It would be useful to compare the sensitivity/resolution possibly selectivity ethanol versus humidity of the investigated perovskite sensor with other sensor approaches e.g. MOX technology. This helps to classify the perovskite approach.

Author Response

See the attached file, please.

Reviewer 4 Report

Accept

Author Response

Thank you for your valuable comments and assistance.